

# Regional scaling of annual mean precipitation and water availability with global temperature change

Peter Greve [*1,2], Lukas Gudmundsson[1], and Sonia I. Seneviratne[1]

[1]Institute of Atmospheric and Climate Science, ETH Zurich, Zurich, Switzerland
[2]International Institute for Applied Systems Analysis, Laxenburg, Austria

*Correspondence to:* Peter Greve (greve@iiasa.ac.at)

**Abstract.** Changes in regional water availability belong to the most crucial potential impacts of anthropogenic climate change, but are highly uncertain. It is thus of key importance for stakeholders to assess the possible implications of different global temperature thresholds on these quantities. Using a large subset of climate model simulations from the 5th phase of the Coupled Modeling Intercomparison Project (CMIP5), we derive here the sensitivity of regional changes in precipitation and precipitation minus evapotranspiration to global temperature changes. The simulations span the full range of available emissions scenarios and the sensitivities are derived using a modified pattern scaling approach. The applied approach assumes linear dependencies on global temperature changes while thoroughly addressing associated uncertainties via resampling methods. This allows us to assess the full distribution of the simulations in a probabilistic sense. Northern high latitude regions display robust responses towards a wetting, while subtropical regions display a tendency towards drying but with a large range of responses. Even though both internal variability and the scenario choice play an important role in the overall spread of the simulations, the uncertainty stemming from the climate model choice usually accounts for about half of the total uncertainty in most regions. We additionally assess the implications of limiting global mean temperature warming to values below (i) $2K$ or (ii) $1.5K$ (as stated within the 2015 Paris Agreement). We show that opting for the $1.5K$-target might just slightly influence the mean response, but could substantially reduce the risk of experiencing extreme changes in regional water availability.

## 1 Introduction

Assessing regional changes in mean-annual precipitation, $P$, and precipitation minus evapotranspiration, $P - E$ (often also referred to as water availability), in the context of on-going global warming is of high relevance for a wide-range of socio-economic sectors. Regional differences in $P$ and $P - E$ pose important challenges to farmers, water resources managers, stakeholders and decision-makers and a comprehensive, easily accessible communication and visualization of complex climate model output is necessary to allow for targeted adaptation and mitigation strategies.

The public debate on climate change is usually limited to a debate about global temperature change, which is, however, an abstract measure and does not enable end-users to infer direct implications for regional to local climate change, especially also with respect to hydroclimatological variables (Victor and Kennel, 2014; Seneviratne et al., 2016). However, due to its

---

*E-Mail: greve@iiasa.ac.at



omnipresence in popular climate communication, global mean temperature $T$ could be used as a general measure of climate change and thereby enable a different communication of regional climate impacts to the public: 'The regional change of a climate variable as a function of global warming'. Many studies use approaches following this guideline, with one of the most common techniques used being summarized as 'pattern scaling'.

In this study we follow the tradition of pattern scaling, but introduce a more rigorous, probabilistic assessment of the underlying uncertainties. Common pattern scaling approaches originally have the goal to use a spatial response pattern in a certain variable (e.g. regional temperature, precipitation) that is derived from observational or (usually) from climate model data with respect to global mean temperature or $CO_2$-changes in order to create a large number of additional scenarios (Santer et al., 1990; Mitchell, 2003). In this study we use the "large number of additional scenarios" created by the utilized pattern scaling
technique to estimate the uncertainty distribution of the response pattern in a probabilistic approach. Pattern scaling approaches have been employed in a large number of studies (see e.g. Tebaldi and Arblaster (2014) for an overview), but many common approaches to estimate the response pattern are also subject to an ongoing debate (Tebaldi and Arblaster, 2014; Herger et al., 2015; Kravitz et al., 2016). To estimate the spatial response pattern in mean-annual $P$ and $P - E$, we adapt here a technique based on the assumption that the scaling relationship between local temperature at each gridpoint and global mean temperature
is linear and that the resulting maps of regression slopes could be used as the response pattern (Solomon et al., 2009). Following a more empirical approach without a priori assumptions on the dependency of regional variables on global temperature, it was recently shown that these findings also hold for extreme temperatures and extreme precipitation, mostly independent of emission scenarios (Seneviratne et al., 2016). This approach was further applied and extended in Wartenburger et al. (2017) by using a comprehensive set of hydroclimatological variables, including both mean-annual $P$ and $P - E$. The assessment presented in
this work builds upon this analysis by utilizing a similar data collection to quantify the associated response pattern.

The scaling relationship between global mean $P$ and global warming has also been analysed in previous studies (Andrews et al., 2009; Frieler et al., 2011; Pendergrass and Hartmann, 2012; Fischer et al., 2014; Pendergrass et al., 2015). At global scales, mean precipitation scales positively with global temperature increase (Knutti et al., 2016), but the associated scaling coefficient is still subject to an ongoing debate and might not necessarily follow a linear relationship (Good et al., 2016). It was
further shown that the magnitude of the scaling relationship depends on the emission scenario (Andrews et al., 2009; Frieler et al., 2011; Pendergrass and Hartmann, 2012; Pendergrass et al., 2015), whereas the scaling relationship of extreme precipitation is independent of the emission scenario (Pendergrass et al., 2015; Seneviratne et al., 2016). The scaling relationship between global mean $P - E$ and global warming was, to our knowledge, only assessed in Wartenburger et al. (2017) and more research is needed to evaluate the full range of potential impacts of regional water availability change.

We aim here to develop a methodological framework in order to assess regional changes in mean-annual $P$ and $P - E$ with respect to global warming by using a comprehensive subset of climate models and considering different emission scenarios. We further account for the internal variability of each projection by considering the year-to-year variability of $P$ and $P - E$. This enables us to generate conservative estimates of the uncertainty distribution of the scaling coefficient for $P$ and $P - E$ at every gridpoint and within specific regions.





Another issue addressed within this study is related to the implications of different warming-degree targets on regional $P$ and $P - E$. At the United Nations Climate Change Conference held in Paris in 2015 (COP21), most nations agreed to limit the increase in global mean temperature to values "well below $2K$" and to ideally not surpass a warming of $1.5K$ above pre-industrial conditions. Thereby, previous goals to limit global warming to "only" $2K$ global warming are significantly

intensified. However, this raises the question of potential implications and differences between these "warming-degree targets" with respect to changes in many other climate variables besides the (rather abstract) value of global mean temperature and especially at regional scales (Seneviratne et al., 2016; Schleussner et al., 2016; Guiot and Cramer, 2016; James et al., 2017). The framework developed within this study allows us to directly assess regional changes in $P$ and $P - E$ in the context of these warming-degree targets, thereby providing important and useful information to decision makers, farmers, water resources

managers, stakeholders and the general public within a specific region.

First, we introduce the climate model data that is utilized within this study before describing the methodological approach that is used to estimate the uncertainty distribution of the scaling coefficients of $P$ and $P - E$ with respect to global warming (Sec. 2). We provide in the following illustrations of the median and the range of the scaling coefficients (Section 3). Next, we comprehensively assess the uncertainty that is stemming from the choice of emission scenario (Section 3.1) and how other

sources of uncertainty contribute to the total uncertainty (Section 3.2). We further apply the new framework to analyse changes between different warming degree targets (Section 4) and summarize and discuss our results also within the context of previous assessments (Sec. 5).

## 2  Scaling - Data and Methodology

The Coupled Model Intercomparison Project, version 5 (CMIP5) ensemble (Taylor et al., 2012) includes climate model pro-

jections forced by four Representative Concentration Pathway (RCP) emissions scenarios (Moss et al., 2010). These scenarios correspond to their relative radiative forcings reached by the end of the 21st century with respect to the preindustrial period: 2.6, 4.5, 6.0, and $8.5Wm^{-2}$ (from hereon referred to as RCP2.6, RCP4.5, RCP6.0 and RCP8.5). We use a total of 14 climate models selected based on prerequisites provided in Fischer et al. (2014). Please note that not all climate models provide data for all emission scenarios (see Table 1). We consider a time period of 120 years beginning in 1980 and ending in 2099 compris-

ing historical simulations for the first 25 years which emerge into simulations of the respective emission scenarios from 2005 onwards. The first 20 years (1980-1999) are used as a common baseline period and values in in mean annual $P$ and $P - E$ are assessed in relative terms [%] with respect to the baseline period. Please note that in case $P - E < 0$ for the majority of models and scenarios, those gridpoints were neglected.

For each model $m$ and each emission scenario $s$ within the 100-year period 2000-2099 ($yr$), these relative values of precip-

itation, $P_{m,s,yr}$, and precipitation minus evapotranspiration, $(P - E)_{m,s,yr}$, are regressed at each gridpoint (or averaged over





a certain region) against mean-annual global temperature, $T_{m,s,yr}$). We use a least squares fit to estimate the parameters of the linear equation:

$$P'_{m,s,yr} = r_{m,s} \cdot T_{m,s,yr} + I_{m,s} \qquad (1)$$

with $r_{m,s}$ denoting the regression slope and $I_{m,s}$ the intercept (and likewise for $(P-E)_{m,s,yr}$). The slope itself provides us

with an estimate of the regional scaling coefficient of P against global changes in T.

Given the annual residuals $R_{m,s,yr} = P_{m,s,yr} - P'_{m,s,yr}$, the uncertainty of the regression slope $r_{m,s}$ is assessed by resampling years $yr'$ of the residuals ($R_{m,s,yr'}$) and fitting the regression slope against the new pairs ($T_{m,s,yr}$,$P'_{m,s,yr} + R_{m,s,yr'}$). Repeating this approach 1000-times at each gridpoint (or within each specific region) provides us with a comprehensive uncertainty measure $\epsilon_{m,s}$ of each model- and scenario-specific regression slope $r_{m,s}$ for both $P$ and $P-E$. We like to point out

that the uncertainty estimated through resampling residuals results in very similar results as computing the uncertainty through using different realisations of a single model (not shown).

This approach allows us to distinguish between three different sources of uncertainty. As illustrated in the conceptual Fig. 1, these are (i) internal variability, representing the uncertainty stemming from interannual variability for each model under each scenario, (ii) the model uncertainty, related to the uncertainty across all models and for a specific scenario (e.g. in terms of

variance: $\sigma = 1/n \sum_{m=1}^{n}(r_{m,rx} - \mu_r)^2$, with $n = 14$ models and $\mu_r$ denoting the average of all scaling coefficient) and (iii) the scenario uncertainty, related to the uncertainty across all scenarios and for a specific model or the multi-model mean (e.g. in terms of variance: $\sigma = 1/n \sum_{rx=1}^{n}(r_{m,rx} - \mu_r)^2$, with $n = 4$ scenarios and $\mu_r$ denoting the average of all scaling coefficient). The total uncertainty denotes the uncertainty across all models, all scenarios and considering the internal uncertainty. Please note, that this approach of attributing uncertainties is very simplistic and neglects any potential relationship between the

individual sources of uncertainty, but is suitable and useful to provide a general measure of the underlying uncertainty sources.

## 3    Scaling - Results

Considering the total uncertainty across all models and scenarios and by additionally including the internal variability, we are able to estimate the uncertainty distribution of the regional scaling coefficient of $P$ ($P-E$) against globally-averaged $T$. Displayed in Fig. 2 are the median and the 10th and 90th quantile of the uncertainty distribution of the scaling coefficient for

both $P$ and $P-E$ at each gridpoint. Positive (negative) values denote an increase (decrease) in $P$ ($P-E$) with increasing $T$. The median scaling coefficient shows positive values in most parts of the northern high latitudes and Asia, but also in eastern Africa for both $P$ and $P-E$. Negative values are found in the Mediterranean region, southern Africa, Australia and in parts of West Africa, as well as Central and South America. Comparing the 10th and 90th quantiles of the uncertainty distribution shows the range of possible scaling coefficients. This range does, in most regions and especially for $P-E$, include the zero coefficient,

which means that the median response is not significant in classical sense ($p = 0.1$) and that there is a non-negligible chance of the median response switching signs. The range is further generally much larger for $P-E$, pointing towards overall higher



uncertainties in the estimation of the scaling relationship. The range is especially large in most subtropical regions (e.g. Sahara, Arab peninsula, India, Australia, etc.). Regions showing a significant increase of $P$ and $P-E$ with global warming are located mainly in the northern high latitudes.

Fig. 3 summarizes these findings by qualitatively showing the probability of experiencing either a positive or negative scaling response in $P$ with respect to global warming. A very likely increase ($90-100\%$ probability) in regional $P$ with ongoing global warming is hence found only within regions of the northern high latitudes, whereas a likely increase ($66-100\%$ probability) is located also in many parts of Asia and North America and to a minor extent also in some regions of South America and Africa. A likely decrease is located in most parts of the Mediterranean region, southern Africa, northeastern South America, Central America and along the Australian coastal regions. A very likely decrease is rarely found only in South Africa. Most other regions show either uncertain or no change. Fig. 3 also shows illustrations of the uncertainty distribution of the scaling coefficient as a function of global temperature increase for a comprehensive subset of SREX regions (Seneviratne et al., 2012) as outlined in the map (see also Table 2 for more information). Very certain responses within the SREX-regions are only found for those in the northern high latitudes (ALA, NAS, NEU), while most other regions show a large spread of the uncertainty distribution (especially e.g. in NEB, NAU).

Similarly for $P-E$, Fig. 4 displays a very likely increase ($90-100\%$ probability) in regional $P-E$ with ongoing global warming for an even smaller portion of land in the northern high latitudes, whereas a likely increase ($66-100\%$ probability) is located throughout the northern high latitudes and similarly to $P$ in many parts of Asia and North America and to a minor extent also in some regions in South America and Africa. A likely decrease is located in parts of the Mediterranean region, southern Africa, northeastern South America, Central America and some parts of Australia. A very likely decrease is only found for single gridpoints, primarily in Central America. Most other (and when compared to $P$ an even higher number of) regions show either uncertain or no change. Fig. 4 also shows illustrations of the uncertainty distribution of the scaling coefficient of $P-E$ as a function of global temperature increase for the same set of SREX regions as shown in Fig. 3. Very certain responses are again only found in the northern high latitudes (ALA, NAS) and southern Asia (SAS), while most other regions show a very large and even larger spread of the uncertainty distribution when compared to estimates of $P$.

## 3.1 Scenario uncertainty

The probability of experiencing an increase/decrease in regional $P$ and $P-E$ with global warming depends on the emission scenario. At global scales, mean precipitation scaling was shown to depend on the emission scenario (Andrews et al., 2009; Frieler et al., 2011; Pendergrass and Hartmann, 2012; Pendergrass et al., 2015), whereas the scaling of extreme precipitation is independent of the emission scenario (Pendergrass et al., 2015; Seneviratne et al., 2016). Here we assess the dependence of regional changes in $P$ and $P-E$ on the emission scenario by analysing the uncertainty distributions of the scaling coefficient for each scenario individually. A conceptual representation of the probability of the scaling coefficient being postive/negative is displayed for $P$ in Fig. 5 and for $P-E$ in Fig. 6 (similar to the total uncertainty as shown in Fig. 3 and Fig. 4). In general, the fraction of regions showing either likely or very likely changes is increasing with the emission scenario for both $P$ and $P-E$, pointing towards a larger uncertainty in the estimation of the scaling coefficient in case the climate change forcing is





weak (RCP2.6, RCP4.5). Further, regions showing very likely changes are much more common under high emission scenarios (RCP6.0, RCP8.5). The drying response in the Mediterranean region is e.g. not evident when considering the RCP2.6 scenario alone, whereas a very likely decrease is found within the RCP8.5 scenario. In fact, parts of the Mediterranean (e.g. in central Spain) even show a likely increase in $P$ and $P - E$ within the RCP2.6 scenario. On the other hand, a likely drying response in parts of central and northen Australia found in RCP2.6 disappears for higher emission scenarios for $P$, or even turns into a wetting response for $P - E$. Robust signals are, again, found in most parts of the northern high latitudes, showing a (very) likely increase across all emission scenarios. A (very) likely decrease is further found for parts of southern Africa and parts of the Amazon region.

A more detailed look on the underlying uncertainty distributions for $P$ ($P - E$) within each SREX-region is provided in Fig. 7 (Fig. 8). It is clearly evident that the uncertainty is largest for low emission scenarios throughout all regions and in most cases lowest for the RCP8.5 scenario (with overall larger uncertainties in $P - E$, please note the different y-axis scales). Additionally, the higher emission scenarios are usually enclosed by the low emission scenarios and the uncertainty is narrowing down towards a more certain signal. However, there are partly huge differences regarding the median response and the location and shape of the uncertainty distribution of a particular emission scenario with respect to other emission scenarios. This is especially evident when comparing low to high emission scenarios. Most prominently, for many regions (e.g. WNA, CAM, MED, WAS, CAS, please see Table 2 for more information on the acronyms) the uncertainty distribution of the RCP6.0 or RCP8.5 scenario is located mainly within the lowest tercile of the RCP2.6 scenario, leading to a dryer response in $P$ ($P - E$) with global warming for high emission scenarios. This finding is, however, reversed in a few other regions (especially NAU and for $P - E$, to a certain extent also in ALA, EAF, SEA). The shapes of the uncertainty distributions are also different between regions and emission scenarios. While the distributions for the low emission scenarios are, in most cases, unimodal, there are bimodal distributions in many (e.g. NEU, WAS, CAS) and even multimodal distributions in a few other regions (e.g. AMZ, NAS) for the higher emissions scenerios (especially RCP8.5). Please note, however, that not all models provide data for the RCP2.6 and RCP6.0 emission scenarios, which might also causes differences between those and the other scenarios (see Table 1 for more information).

## 3.2 Sources of uncertainty

Besides the scenario uncertainty we also introduced two other sources of uncertainty in Sec. 2, the internal variability and the model uncertainty which contribute to the total uncertainty. Here we assess the fraction of uncertainty which each source contributes to the total uncertainty. We follow the approach of Hawkins and Sutton (2009), which was also adapted in Orlowsky and Seneviratne (2013). Therefore we compare (i) the average over the variances of the uncertainty distributions of each model under each emission scenario (internal uncertainty), (ii) the average of the variances of scenario-specific uncertainty distributions of each model (model uncertainty) and (iii) the variance of the averages of all uncertainty distributions within a specific scenario (scenario uncertainty). Even though this approach of attributing uncertainties is very simplistic (see section 2), it provides basic information on the composition of different uncertainty sources within the total uncertainty. The percentage of the total uncertainty that stems from a particular source is illustrated for both $P$ and $P - E$ in Fig. 9. For all SREX regions,



there is generally no uncertainty source which is significantly dominating the total uncertainty. However, in most regions the largest source of uncertainty stems from model uncertainty, which is contributing up to ca. 3/4 of the total uncertainty in some regions and is especially large in most northern high latitude regions (CGI, NEU, NAS, except ALA). Internal variability contributes between $20 - 40\%$ to the total uncertainty, with highest values found for contrasting regions such as ALA, CEU

and SAU. Internal variability seems to be rather low in AMZ, EAS and NAU for $P$ and in NEB, WAF and SAS for $P - E$. Scenario uncertainty contributes between ca. $5 - 30\%$ to the total uncertainty with those regions reaching highest values that have differing locations of the uncertainty distributions between low and high emission scenarios as shown in Fig. 7 and Fig. 8 (especially WNA, ENA, CAM, MED, WAS, CAS, and NAU). It is further interesting to note that scenario uncertainty is generally lower and internal variability generally larger for $P - E$ when compared to $P$. However, even though the scenario

uncertainty is the overall weakest source of uncertainty in most regions, it is by no means negligible. Please note, again, that the scenario uncertainty interferes especially with the rather large model uncertainty and we do not account for such relationships in this approach.

## 4   Application: Assessing warming degree limits

At the United Nations Climate Change Conference held in Paris in 2015 (COP21), most nations agreed to limit the increase

in global mean temperature to values well below the previously set goal of $2K$ and to limit warming to not more than $1.5K$ above pre-industrial conditions. However, global mean temperature is an abstract value and provides no information about direct implications at regional scales and with respect to other climate variables such as regional, mean-annual $P$ or $P - E$. The framework developed within this study enables us to directly assess the regional response of $P$ and $P - E$ to these targets and to study differences between them. Using the uncertainty distributions of each SREX region and scaling them to either

$1.5K$ or $2K$ (as illustrated in Fig. 10 for $P$ and in Fig. 11 for $P - E$) allows us to study differences both in the median response as well as in the tails of the distribution. It is, however, naturally evident that in regions with a weak median scaling response the difference between the warming degree targets is small regarding the median response itself (e.g. AMZ, CEU, WAS), whereas in regions with a stronger median scaling (e.g. ALA, NEU, NAS) an additional $0.5K$ warming could lead to substantial differences. Nonetheless, even though the difference in the median might be small, the difference in the tails

of the uncertainty distributions are in most cases significant and stress an increased risk of experiencing strong changes in $P$ and $P - E$. As an example for the Mediterranean region (MED): The median responses of $P$ to $1.5K$ global warming vs $2K$ global warming are not strongly different, while there are stronger differences at the tails, showing that the 1.5K limit would avoid a decrease of P of more than $20\%$, which can on the other hand not be excluded with the 2K limit. This behavior is even more evident for $P - E$ and also occurs in regions with almost no median response (e.g. CEU). Also the irregularity of

the distribution further amplifies this behavior in some regions. In summary, opting for a low warming degree target (such as $1.5K$) might just slightly influence the mean response but could substantially reduce the risk of experiencing more extreme changes in regional $P$ and $P - E$.





## 5 Conclusions

We developed here a framework building upon the pattern scaling approach to assess regional changes in mean-annual $P$ and $P - E$ with respect to global mean $T$-increase by utilizing a comprehensive subset of climate models and considering all available emission scenarios. We further took into account internal variability from each projection by accounting for the

5 year-to-year variability of $P$ ($P - E$). This enabled us to assess a conservative estimate of the uncertainty distribution of the scaling coefficient of $P$ ($P - E$) to global warming at every gridpoint or within SREX regions.

Analysing maps of the median response and the responses in the 10th and 90th quantile of the gridpoint-specific uncertainty distributions showed low uncertainties and positive scaling coefficients (thereby a certain increase in $P$ and $P - E$ with global warming) within most northern high latitude regions. Slight decreases in the median response together with large uncertainties

(and thereby an uncertain decrease in $P$ and $P - E$ with global warming) are found for most subtropical regions. Uncertainties are, however, larger for estimates of $P - E$ and do hence not permit robust conclusions for many regions. Our results support previous findings of hydroclimatological changes (Greve and Seneviratne, 2015), but provide a new, probabilistic and rigorous perspective on the assessment of uncertainties in regional hydroclimatological changes under conditions of ongoing global warming and extent the wealth of studies investigating pattern scaling approaches of climate variables (Tebaldi and Arblaster,

2014; Herger et al., 2015).

Assessing scenario-specific uncertainty distributions revealed strong regional differences between different emission scenarios. It is evident that weaker climate change signals within the low-emission scenarios (RCP2.6, RCP4.5) lead to high uncertainties in the estimation of scaling coefficients. A very likely change in regional $P$ only emerges under high emission scenarios (RCP6.0, RCP8.5) and is even less likely to occur for $P - E$. In some regions low emission scenarios further show

likely opposite changes compared to changes identified in higher emission scenarios (both switching from a likely wetting to a very likely drying in parts of MED, or from a likely drying to no change in $P$ or even a likely wetting in $P - E$ in NAU). A closer look a the uncertainty distributions shows large differences both in location and shape across regions and emission scenario. However, in most cases higher emission scenarios point towards a dryer response than low emission scenarios (with a few regions showing, however, the opposite behavior).

This led us to the analysis of the relative contribution of single sources of uncertainty to the overall uncertainty. It is shown that model uncertainty is largest in most regions, but is not significantly dominating the overall uncertainty. It is therefore important that both internal variability and scenario uncertainty are considered as well in order to get a complete picture of the total uncertainty. Comparing mean annual $P$ and $P - E$ shows that scenario uncertainty is generally lower and internal variability generally larger for $P - E$.

We further assessed the implications of different warming degree limits on changes in regional $P$ and $P - E$. At the COP21, most nations agreed to limit the increase in global mean temperature to values well below the previously set goal of $2K$ and to consider limiting warming to not more than $1.5K$ above pre-industrial conditions. Comparing these two targets reveals naturally little differences in the mean response in regions where the mean response is small anyway. However, since uncertainties are large, especially for $P - E$, there is a nonlinear increase in the risk of experiencing more extreme changes. Therefore,





opting for a low warming degree target (such as $1.5K$) might just slightly influence the mean response but could substantially reduce the risk of experiencing extreme changes in regional $P$ and $P - E$. This means that even though the discussion about the implications of $1.5K$ vs. $2K$ global warming might be moot for the mean response, it is, given the underlying, large uncertainties of climate projections, absolutely necessary to more closely investigate the potentially large increase in the risk

of experiencing extreme change. This is especially important in order to enable robust decision making to ensure adequate development pathways and to avoid the risk of maladaptation; in the specific case of changes in mean-annual $P$ and $P - E$ this is e.g. of high relevance for water resources managers and farmers.

*Data availability.* We acknowledge the World Climate Research Programme's Working Group on Coupled Modelling, which is responsible for CMIP, and we thank the climate modelling groups for producing and making available their model output. For CMIP the U.S. Department

of Energy's Program for Climate Model Diagnosis and Intercomparison provides coordinating support and led development of software infrastructure in partnership with the Global Organisation for Earth System Science Portals. The data used in this study are available through the Coupled Model Intercomparison Project Phase 5 at http://pcmdi9.llnl.gov/esgf-web-fe/

*Competing interests.* The authors declare no competing financial interests.

*Acknowledgements.* We thank Jakob Zscheischler for providing useful input regarding methodological aspects of this study and Urs Beyerle

and Jan Sedlacek for processing the CMIP5 data.



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

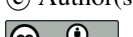


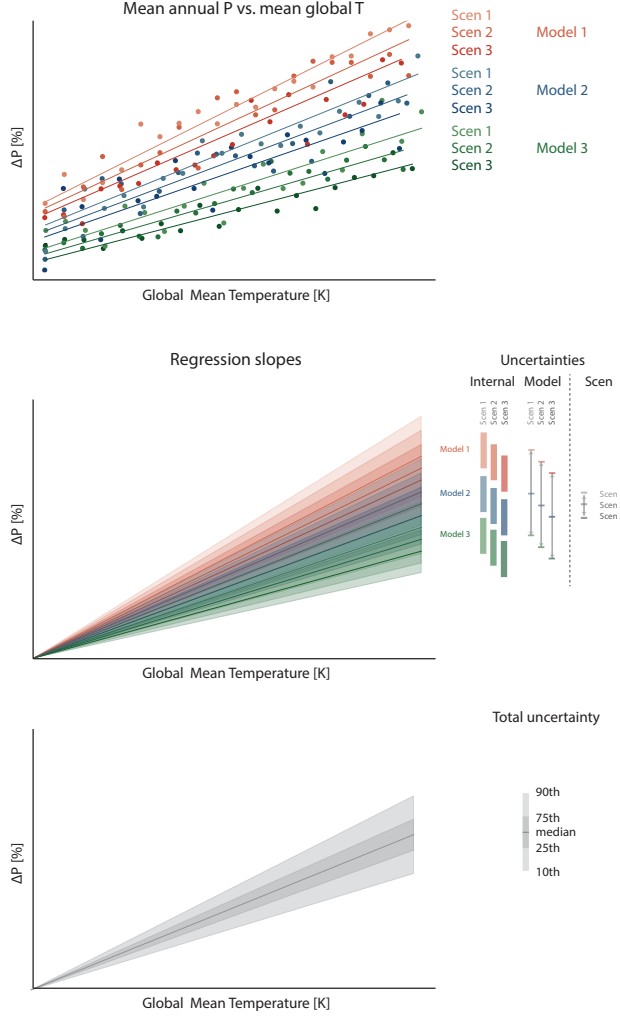

**Figure 1.** Conceptual illustration of deriving the uncertainty distribution of the scaling coefficient of $P$ with respect to global warming from multiple models forced by multiple emissions scenarios. (Top) For each model under each scenario we regress the relative change in mean-annual $P$ ($\Delta P$, with respect to a baseline period) against global mean $T$. We thereby obtain the regression slope which is the scaling coefficient of $P$ to global warming. The year-to-year variability further causes the estimate of the slope to be uncertain. We account for this uncertainty by numerically estimating the uncertainty distribution of each model- and scenario-specific regression slope through resampling the residuals in a bootstrapping approach. (Middle) This uncertainty is associated with every model run and represents the internal variability. The average of the uncertainties stemming from the range of all individual models within a certain scenario represents the model uncertainty and the uncertainty associated with the range of all scenario-specific multi-model means represents the scenario uncertainty. The uncertainty distribution is illustrated here as a function of global temperature increase. (Bottom) The Total uncertainty combines all sources of uncertainty and provides a conservative estimate of regional $\Delta P$ as a function of global warming, that can be used to assess either the median response or to study changes in any other quantile of the uncertainty distribution.



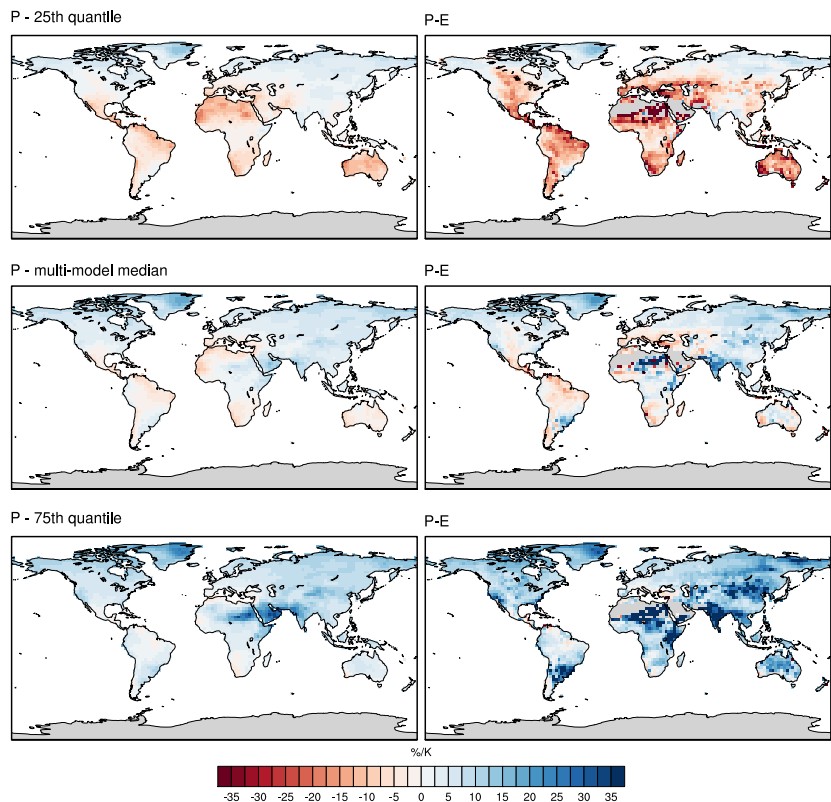

**Figure 2.** Median (middle), 10th (top) and 90th quantile (bottom) of the sensitivity of $P$ (left) and $P - E$ (right) to changes in global mean temperature $[\%/K]$. A total of 14 CMIP5 models and all scenarios (RCP2.6, RCP4.5, RCP6.0, RCP8.5) are considered.





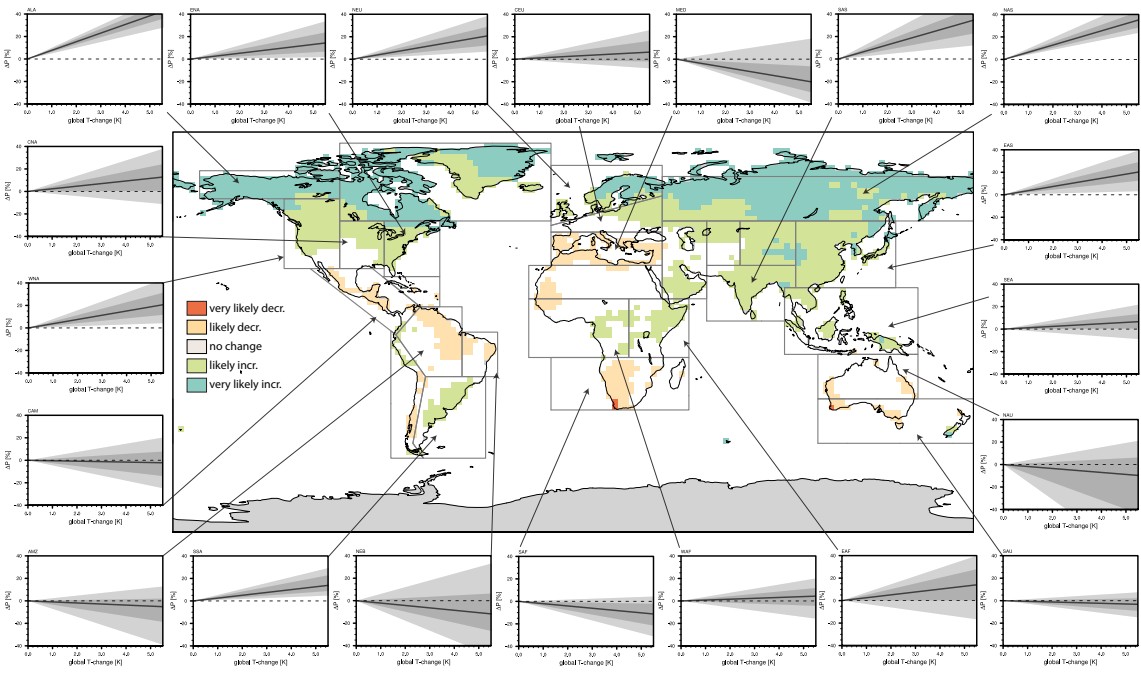

**Figure 3.** Conceptual summary of the probability that the slope of $P$ is negatively/positively different from zero considering all climate models and all scenarios. Panel plots illustrates the uncertainty distribution of the sensitivity of $P$ to global temperature change as a function of global mean temperature change averaged for each SREX regions outlined in the map (the shading in each panel plot corresponds to those illustrated in Fig. 1).





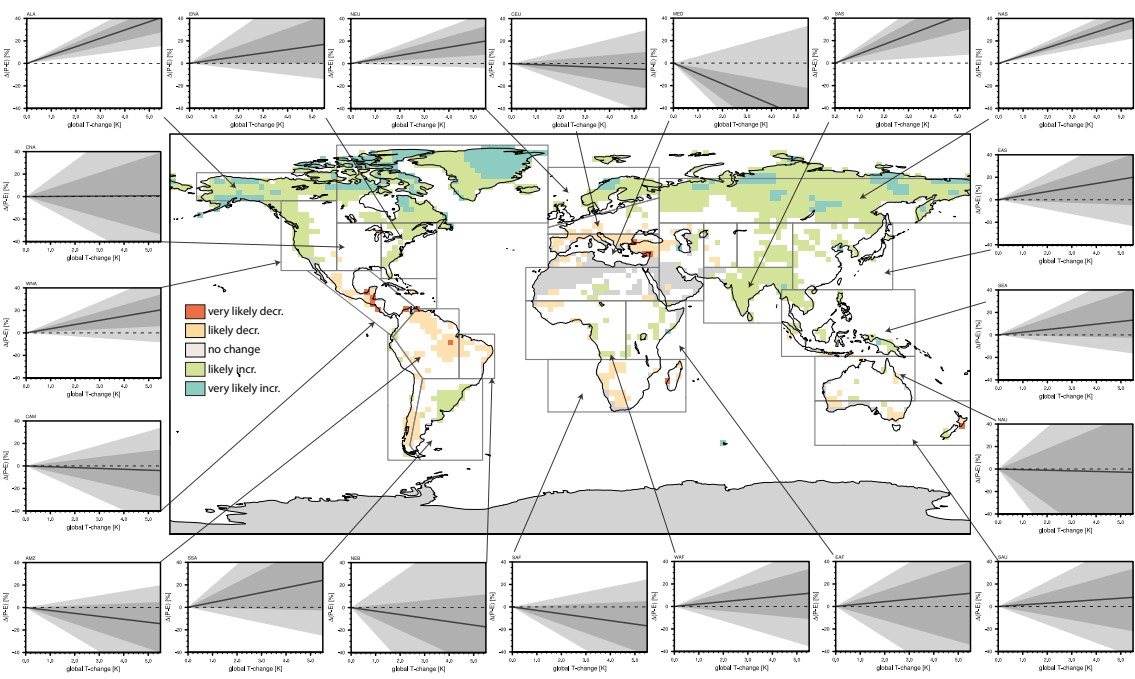

**Figure 4.** Conceptual summary of the probability that the slope of $P - E$ is negatively/positively different from zero considering all climate models and all scenarios. Panel plots show the uncertainty distribution of the sensitivity of $P - E$ to global temperature change as a function of global mean temperature change averaged for each SREX regions outlined in the map (the shading in each panel plot corresponds to those illustrated in Fig. 1).

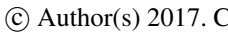



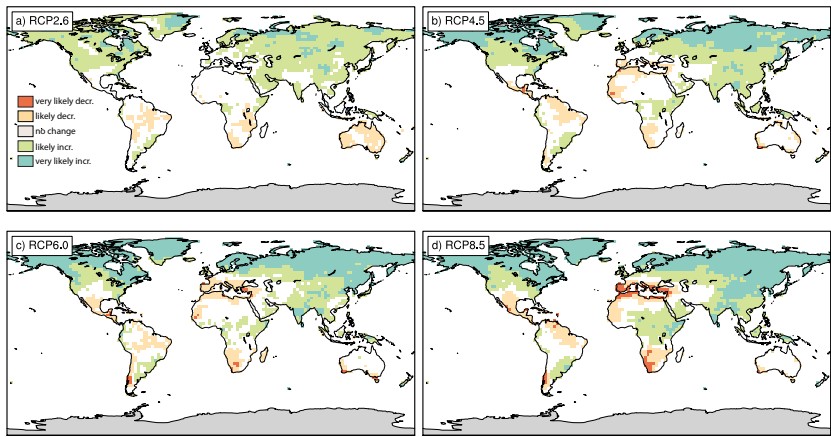

**Figure 5.** Conceptual summary of the probability that the slope of $P$ is negatively/positively different from zero considering all climate models and a) the RCP2.6, b) the RCP4.5, c) RCP6.0 and d) RCP8.5 emission scenario only. See Fig. 3 for comparison.



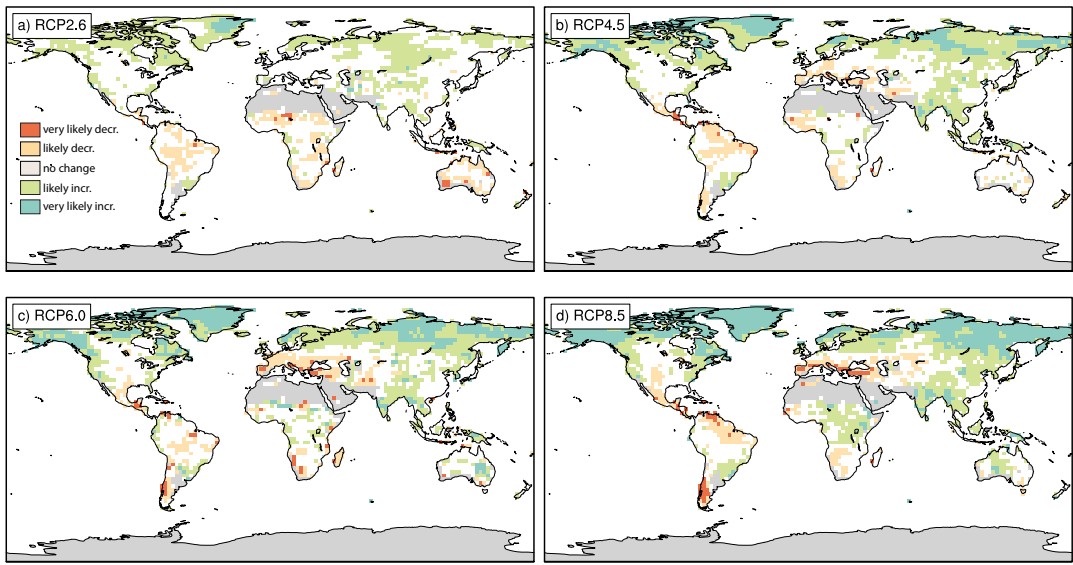

**Figure 6.** Conceptual summary of the probability that the slope of $P - E$ is negatively/positively different from zero considering all climate models and a) the RCP2.6, b) the RCP4.5, c) RCP6.0 and d) RCP8.5 emission scenario only. See Fig. 4 for comparison.



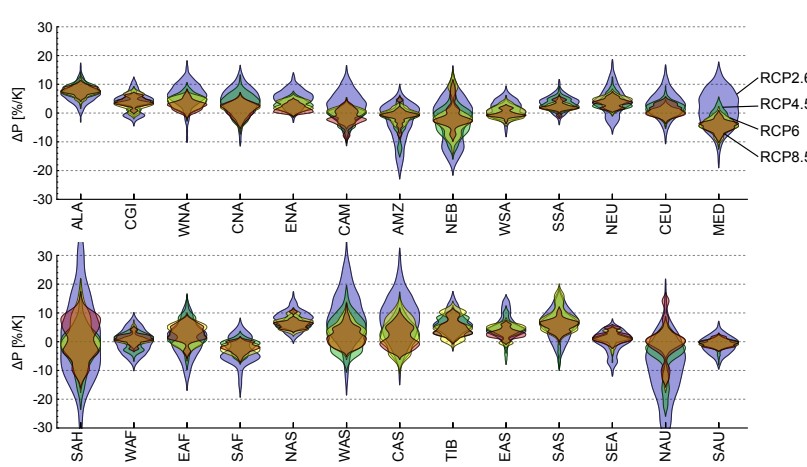

**Figure 7.** Uncertainty distributions (shown as violin plots) of the sensitivity of $P$ to global mean temperature change for each emission scenario averaged over all SREX regions (as outlined in Table 2 and Fig. 3).





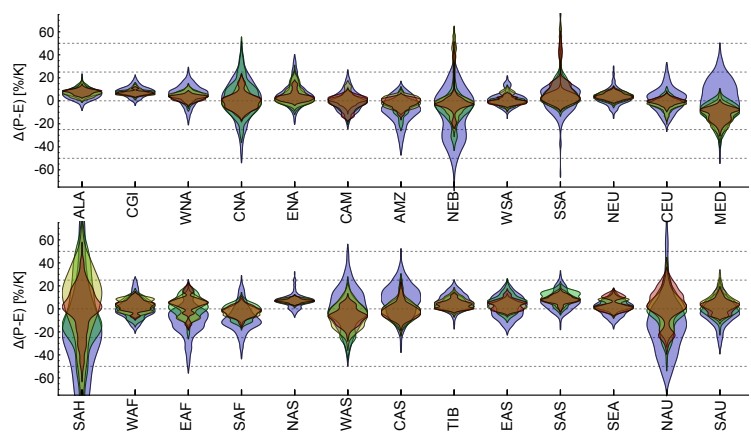

**Figure 8.** Uncertainty distributions (shown as violin plots) of the sensitivity of $P - E$ to global mean temperature change for each emission scenario averaged over all SREX regions (as outlined in Table 2 and Fig. 4). It is important to note that the data considered to estimate the area average is scarce in some regions (e.g. SAH). Please also refer to Fig. 7 for more information.



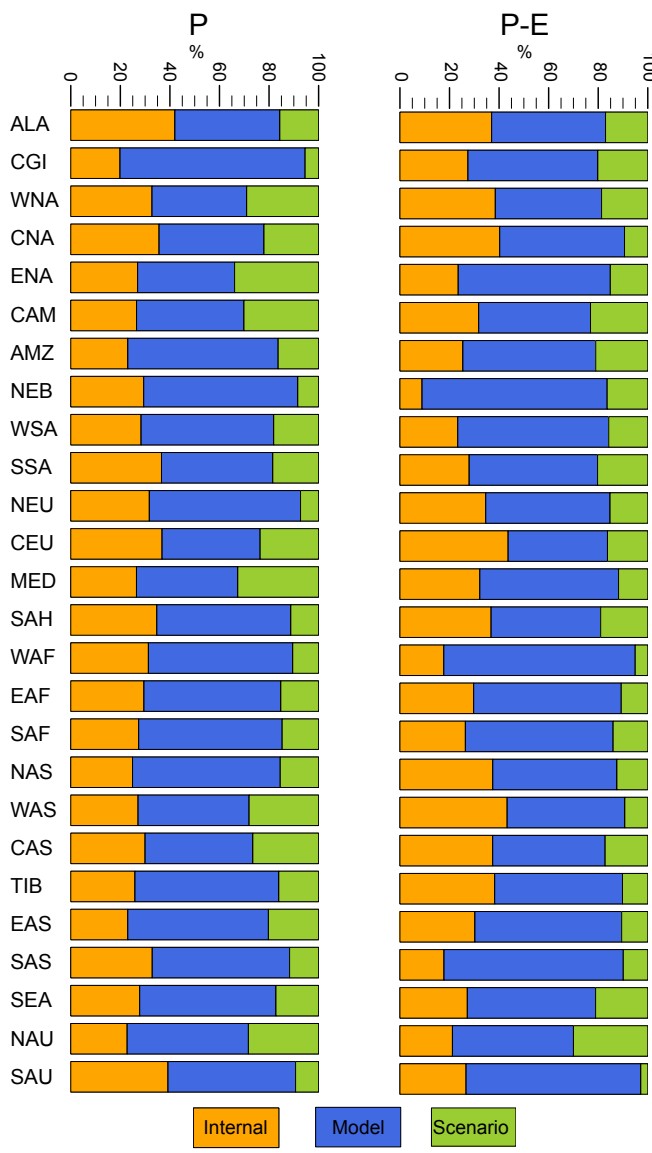

**Figure 9.** Sources of uncertainty in the sensitivity of $P$ (left) and $P-E$ (right) to global mean temperature change averaged over each SREX regions as outlined in Fig. 3.



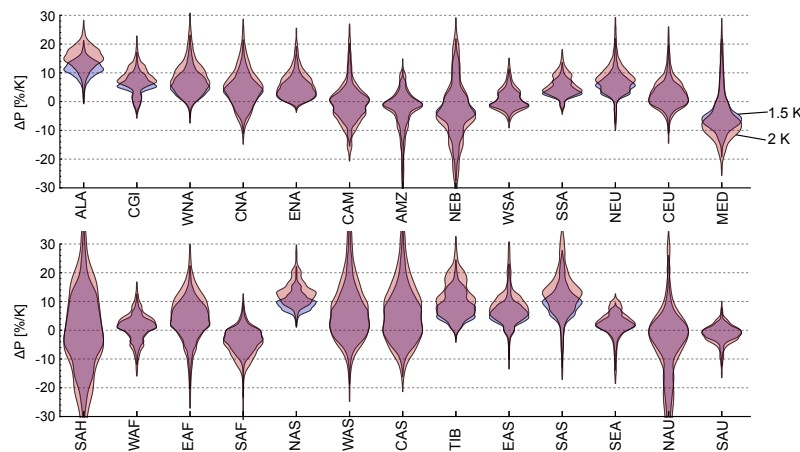

**Figure 10.** Uncertainty distributions (shown as violin plots) of the sensitivity of $P$ to global mean temperature change for two different degrees of global mean temperature change, which correspond to the widely used warming degree limits of $1.5K$ and $2K$. The estimates are averaged over all SREX regions (as outlined in Fig. 3 and Table 2).




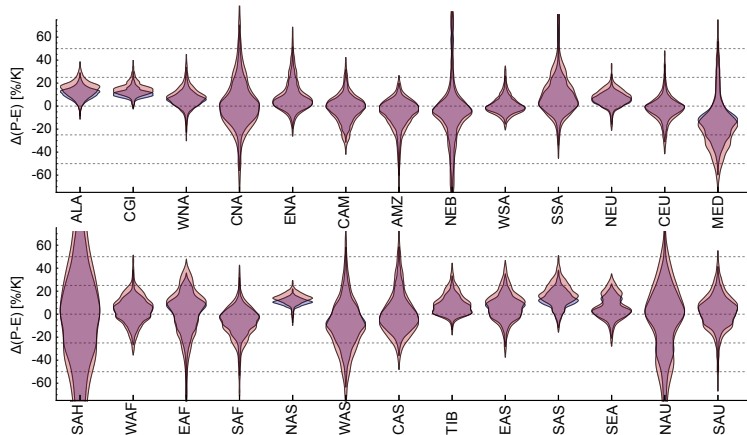

**Figure 11.** Uncertainty distributions (shown as violin plots) of the sensitivity of $P - E$ to global mean temperature change for two different degrees of global mean temperature change, which correspond to the widely used warming degree limits of $1.5K$ and $2K$. The estimates are averaged over all SREX regions (as outlined in Fig. 3 and Table 2). It is important to note that the data considered to estimate the area average is scarce in some regions (e.g. SAH). Please also refer to Fig. 10 for more information.



**Table 1.** List of models and availability under each emission scenario.

| Model | RCP2.6 | RCP4.5 | RCP6.0 | RCP8.5 |
|---|---|---|---|---|
| ACCESS1-3 | | x | | x |
| bcc-csm1-1 | x | x | x | x |
| CanESM2 | x | x | | x |
| CESM1-BGC | | x | | x |
| CMCC-CMS | | x | | x |
| CNRM-CM5 | x | x | | x |
| CSIRO-Mk3-6-0 | x | x | x | x |
| FGOALS-g2 | x | x | | x |
| GISS-E2-R | x | x | x | x |
| HadGEM2-ES | x | x | | x |
| IPSL-CM5A-MR | x | x | x | x |
| MIROC5 | x | x | x | x |
| MRI-GCM3 | x | x | x | x |
| NorESM1-M | x | x | x | x |



**Table 2.** List of Acronyms for all 26 SREX-regions (Seneviratne et al., 2012)

| Region | SREX-Acronym |
|---|---|
| Alaska/NW Canada | ALA |
| Eastern Canada/Greenland/Iceland | CGI |
| Western North America | WNA |
| Central North America | CNA |
| Eastern North America | ENA |
| Central America/Mexico | CAM |
| Amazon | AMZ |
| NE Brazil | NEB |
| West Coast South America | WSA |
| South- Eastern South America | SSA |
| Northern Europe | NEU |
| Central Europe | CEU |
| Southern Europe/the Mediterranean | MED |
| Sahara | SAH |
| Western Africa | WAF |
| Eastern Africa | EAF |
| Southern Africa | SAF |
| Northern Asia | NAS |
| Western Asia | WAS |
| Central Asia | CAS |
| Tibetan Plateau | TIB |
| Eastern Asia | EAS |
| Southern Asia | SAS |
| Southeast Asia | SEA |
| Northern Australia | NAS |
| Southern Australia/New Zealand | SAU |