# Peer review of "Regional scaling of annual mean precipitation and water availability with global temperature change"

_Earth System Dynamics, 2017_

## Referee Comment (RC1) · Anonymous Referee #1 · 23 Jul 2017

This paper applies the regional scaling concept (e.g. Tebaldi and Arblaster, 2014) to the distributions of annual mean precipitation (P) and precipitation minus evaporation (P-E) anomalies. The rationale behind this methodology is based on the work by Seneviratne et al., 2016, in which it is shown that local temperature and precipitation extremes scale linearly with global mean temperatures in CMIP5 CO2-increasing scenarios projections. The mean and uncertainty ranges of P and P-E local responses scale with the global mean temperature in the same way, independently of the emission scenario. This paper extends Seneviratne et al., 2016, considering annual mean local precipitation response scaling with global mean temperatures over all the SREX regions (Seneviratne et al., 2012). The impacts of uncertainty due to models internal

variability, to the inter-model spread and to the scenario spread are also separately accounted for.

**1  GENERAL COMMENTS**

The aim of the paper is focused and clear, and it well suits in the current debate about the regional impact of global temperature change. Exploring the ranges of applicability of the pattern scaling approach allows to improve the capability to communicate the impact of climate change to the stakeholders and public opinion. In this sense the assessment of regional changes in P and P-E is of outmost importance for the adaptation to future changes in local water resources.

The regional pattern scaling is here assessed in terms of a basic least squares fit, regressing annual mean P and P-E anomalies over annual mean global temperature anomalies at every grid-point. The uncertainties related to internal variability, model and scenario-related uncertainties for each model are obtained by resampling the residuals 1000-times over each gridpoint. The empirical probability distribution provides thus a way to characterize the range encompassing the median values of the regression slopes, allowing the distinction between very likely (90-100%), likely (66-100%) increase/decrease in P or P-E. The use of a basic linear scaling is justified by the lack of a-priori information about the shape of the annual mean P and P-E distributions over the various regions is not available. This shifts the focus from the choice of suitable downscaling techniques to the evaluation of uncertainty ranges attributable to the regression coefficients. In this respect, I think that the manuscript partially fails in discussing the impact of models' choice. Seneviratne et al., 2016 outlined limitations to the regional scaling pattern approach in this context. Particularly, point 4) of their discussion emphasized the risk of common biases through models for some regional phenomena. They point out that a careful model evaluation against appropriate observations would be necessary to deal with this problem. The internal variability

of considered model and the multi-model ensemble uncertainty is addressed in the manuscript, but some more effort should be devoted to the evaluation of each model. Particularly, the biases induced by the imbalance in the water mass budget and the impact of different choices of the model ensembles should be carefully addressed, in order to assess the applicability of the method and the robustness of the findings.

**2  SPECIFIC COMMENTS**

l. 22-23 p. 3: if not argued the choice of the models may look a bit arbitrary. On one hand the authors rely on Fischer et al., 2014 to choose only one model for each modelling centre. On the other hand they do not consider the impact of biases in the atmospheric moisture budget. Models are known to show diverse estimates of the global mean water budget (cfr. Liepert and Lo, 2012, Env. Res. Lett.) and this may in principle prevent from consistent estimates of regional changes in P and P-E. Evaluating the long-term mean atmospheric moisture budget in control runs, identifying the regions where climate models diverge from available observations is thus a pre-requisite to this analysis. An inconsistent global mean moisture budget is a potential source of biases and the impact of adding/removing individual models should be carefully evaluated. In the framework of the regional pattern scaling, it would also be relevant to compare the atmospheric moisture budget separately over continents and oceans with the total runoff from the continents (which is a standard output in climate models, if I am not wrong is named as "mrro"), in order to provide a complete description of the hydrological cycle consistency in the model;

l. 23-24 p. 3: as also mentioned in l. 23-24 p. 6 the choice of ensembles with different numerosity is inherently a considerable source of uncertainty, unless one considers the 14-member and 7 (in the case of RCP6.0) and 11 (in the case

of RCP2.6) members ensembles having the same statistical properties. For the same reasons motivating the previous comments, the impact of adding/removing a model from the ensembles should be carefully evaluated. To be on the safe side, I would suggest to reconsider the RCP2.6, RCP4.5 and RCP8.5 scenarios only using those models that are available in the RCP6.0 and discuss about the presence/absence of significance differences in the results. In the occurrence of significant differences I would try to identify and describe those models significantly reshaping the ensemble distribution;

l. 12-18 p. 4: the definition of variances might be clarified by labelling each sigma with a different subscript, either referring to internal variability, model uncertainty, scenario uncertainty;

l. 15 p. 4: following above comment, it should be specified how to deal with the model uncertainty when the ensemble numerosity is lower than 14, e.g. in the RCP6.0 n=7?

l. 30-31 p. 4: to me it is not clear how the authors deal with uncertainty ranges including the zero value for the slope. Could you please expand this statement?

l. 5-6 p. 5: the authors might want to comment on the fact that spatial averaging over northern high latitudes is not the same as spatial averaging at lower latitudes, and this shall be considered when discussing the significance of results at different latitudes. I wonder if one could compare circles of latitude somewhat weighting the likelihood of the changes with the cosine of latitude or the surface area covered by each circle.

l. 19-22 p. 6: the authors mention the different shapes of the uncertainty distributions for different SREX regions in P and P-E regression slopes. Could you please specify whether you refer to the P, P-E or both variables. Otherwise these statements appear a bit arbitrary and one might want to consider removing them;
l.1 p. 7 (and l. 25 p. 8): please specify the meaning of "significantly";

l. 4-6 p. 7: the authors list here a number of SREX regions characterized by larger/smaller internal variability, model uncertainty, scenario uncertainty compared to other regions. I think some more explanation might be welcomed here, rather than just listing the findings over the various regions. Why these regions, rather than others? For instance, the large model uncertainty over northern high latitudes might be related to the more relevant signal ("very likely increase" in precipitation), whereas the large internal variability over the two sides of the Tropical-Northern Pacific might reflect some relatively well understood mechanisms of inter-annual variability, such as the QBO (cfr. Labat et al., 2004, Geophys. Res. Lett.);

l. 14-16 p. 7: repetition of l. 2-4 p. 3, consider removing;

**3  TECHNICAL COMMENTS**

l. 25 p. 3: remove one "in";

l. 17-18 p. 4: replace "coefficient" with "coefficients";

l.23 p. 6: replace "causes" with "cause";

l. 14 p. 8: replace "extent" with "extend;

Table 2: the acronym for Northern Australia should be NAU (instead of NAS);

[Figure]

---

## Referee Comment (RC2) · Anonymous Referee #2 · 26 Jul 2017

In this manuscript, Greve, Gudmundsson, and Seneviratne examine the scaling of local and regional precipitation and P-E with global mean surface temperature in climate change projections. They diagnose the likelihood of increases or decreases with warming in both quantities, and characterize and identify uncertainty due to internal variability, structural model differences, and differences in emissions scenario. To address the impacts of P and P-E on the 1.5 and 2°C warming targets, they quantify the P and P-E responses and their uncertainty in each of a variety of land regions in response to the two targets. They find that the mean changes in P and P-E are indistinguishable for 1.5 and 2°, but that the two warming targets do differ in the tail of risk estimates, with a higher risk of the largest changes for 2°C warming compared to 1.5.

[Figure]

This work makes a useful contribution to the literature, as regional changes in mean precipitation scaling have not yet been diagnosed. The maps and violin plots for individual regions are particularly useful. There are a few issues I think should be addressed to improve the manuscript.

Scientific issues

P3 line 26-27: Why omit locations where P-E<0?

Figures 1, 3, and 4: In all dP versus T plots with the exception of the top panel of Fig. 1, the regressions cross through the origin. The uncertainty in the regression slope is shown as occurring entirely at the upper end of the temperature change axis. These are in conflict with the top panel of Fig. 1, where the regression slopes do not pass through the origin. Internal variability is always present, so we would expect small changes in dP even when dT=0. Is there a better way to visualize the range of regression slopes and their uncertainty? The violin plots are quite useful and do not contain these distortions.

P4 line 28/30: I believe the 10th-90th percentile confidence corresponds to p=0.2, rather than p=0.1. In addition, why do you choose 10th and 90th percentile – since these are wider bounds than is customary? Why not 5 and 95 (p=0.1), or 2.5 and 97.5 (p=0.05)?

P4 line 10, P6 line 28: The methodology of Hawkins and Sutton (2009) assumes that variance is constant over the course of simulations. They only examined temperature, for which this assumption is more or less valid. It seems to me that resampling residuals would rely on the same assumption. For precipitation, it is not the case that precipitation variability is generally constant – instead, it increases in most regions (e.g., Räisänen, 2002). Do you think increasing precipitation variability would affect your uncertainty decomposition, and if so, how?

Typos and grammatical comments
P2 line 31: "comprehensive subset": This is contradictory, since a subset is by definition not comprehensive.

P5 line 9: "A very likely decrease is rarely found only in South Africa." I think what you mean is that a decrease with very likely confidence is found only in South Africa, and therefore it is rare; your wording means something else: that very likely decrease is often found in many places, rarely only in South Africa.

P6 line 12-14: "the higher emission scenarios are usually enclosed by the low emission scenarios and the uncertainty is narrowing down"; "partly huge differences": These phrases are not quite grammatically correct.

Fig. 1: "Global mean Temperature" should probably be "Global mean Temperature Change"

References

Räisänen, J.: COâĆĆ-Induced Changes in Interannual Temperature and Precipitation Variability in 19 CMIP Experiments, J. Clim., 15, 2395–2411, doi:10.1175/1520-0442(2002)015<2395:CICIIT>2.0.CO;2, 2002.

---

## Referee Comment (RC3) · Anonymous Referee #3 · 27 Jul 2017

The authors investigate regional changes in precipitation (P) and water availability (expressed in terms of precipitation minus evaporation, P-E) as a function of global temperature changes in a sub-set of the CMIP5 simulations. They further decompose the uncertainties by sources related to climate variability, scenario, and model choice. They find robust changes towards wetting in northern high-latitude regions, and tendencies towards drying in subtropical regions, however associated with larger uncertainties. In particular, they also discuss changes related to political global warming limits of 1.5K and 2K.

This study is a worthwhile contribution to the literature, addressing the relevant topic

of regional impact-relevant responses related to different amounts of global warming. The manuscript is mostly well written, but some clarification is needed at a few places. I also have a few more major questions related to the methodology, but think that it should be possible to clarify these with some revisions.

**Major comments:**

(1) The authors use resampling to estimate the effects of internal climate variability. They mention that this leads to similar results as using different realisations of one model but do not show results. As estimation of different uncertainty sources, including variability, is one of the main goals of this paper, I think the authors should provide evidence that their approach by just resampling results from one run does actually lead to comparable results to analysing different runs. This seems important as usually effects of variability are estimated from a number of runs started from different climate states with respect to internal variability.

(2) The authors document some larger differences in the response between different scenarios, and seem to discuss these differences in the context of different strength of the GHG forcing. However, also the aerosol concentrations differ between the different RCP scenarios, and I wonder to which extent these scenario differences of P and P-E changes could be attributed to differences in aerosols?

**Specific and technical comments:**

- Abstract, line 3: I'd remove "large" as I don't think 14 model simulations is a "large" sub set of the total number of runs available in CMIP5

- Abstract, line 6: (Please also check throughout the text!) I suggest avoiding "dependency" when discussing the relationship of regional climate with global mean temperature. Better just say "linear relationship" here.

- page 1, line 21: suggest adding "public and political debate"

- page 2, line 14: I wonder if the assumption of a linear relationship is justified when

investigating changes at individual grid cells from individual ensemble members. Especially P and P-E can be rather noisy variables, strongly affected by low-frequency variability, so it might help to justify the robustness of the approach if the authors provided some tests whether linearity is a reasonable assumption in this context.

- page 2, line 20: This sentence is ambiguous, it seems like "this work" and "this analysis" refer to different studies, but it is not really clear what refers to what.

- page 3, line 1: insert "global" warming-degree targets

- page 3, line 23: many readers may not be aware what exactly the "prerequisites provided in Fischer et al." are – for better readability please briefly summarise

- page 3, line 25: the historical runs include the year 2005, therefore if starting in 1980 this should be "26 years" and "from 2006 onwards".

- page 3, line 26: remove duplicate word "in"

- page 3, line 26/27: Sippel et al (https://doi.org/10.5194/hess-21-441-2017) discuss that assessing changes relative to a short reference period may lead to bias in the out-of-reference period. As the authors chose here to use only 20 years as baseline, I am wondering whether their quantifications of changes would be affected by such biases?

- page 3, line 27: Sentence not clear, does "majority of models" suggest that some models are treated different than others?

- page 4, line 1: What kind of least squares fit did you use, e.g. ordinary or orthogonal (i.e. minimising squared differences only in y-direction or in both x and y-directions)? I think there may also be some error in the T values, so orthogonal least squares might be most appropriate?

- page 4, line 24: Please check, is it 10th and 90th quantile, or 25th and 75th quantile as written on Figure 2?

- page 4, line 25: Very confusing use of parentheses. If I follow your logic that the text in

parentheses indicates some opposite results/statements then this sentence seems to suggest negative values always relate to (P-E) – which of cause is nonsense. Please also see this text by A. Robock (https://eos.org/opinions/parentheses-are-are-not-for-references-and-clarification-saving-space), and consider rewriting this sentence in a more readable (and clearer!) way.

- page 5, line 5: based on only 14 models, the 90-100

- Page 5, line 21: Figure 4 shows the scaling relationships as regression lines, it does not explicitly show the "coefficients" as claimed in this sentence.

- page 5, line 26: replace "/" by "or"

- page 6, line 1: remove "much"

- page 6, line 3: replace "within" with "in", and "parts" with "individual grid cells"

- page 6, line 4: replace "within" with "in"

- page 6, line 7: consider adding the clarification "(very) likely decrease [across all scenarios]".

- page 6, line 21: replace "many" with "some"

- page 6, line 25: The structure of the Results section seems somewhat confusing. Section 3.2 is named "sources of uncertainty" – but didn't already section 3.1 discuss one specific source of uncertainty? Please consider restructuring Section 3 more logically, or at least use more suitable sub-section names, e.g. "3.2 Comparing different sources of uncertainty".

- page 7, line 26: add "with stronger global warming" (or similar) at the end of the sentence after "P-E".

- page 9, line 1: This sentence is a literal repetition of page 7, line 30-32. Please consider rephrasing one of these instances. Otherwise this is a very nice conclusion.

- Figure 2: It looks like ocean areas and Antarctica were removed. This is not explicitly mentioned in the text – are these regions removed due to P-E<0 here (page 3, line 27) ? As mentioned above, please also check for consistency whether it is 10th/90th percentile (or if you wish to express in quantiles: 0.1 and 0.9), or 25th/75th

- Figure 3/4: The text and labelling of the T-P scaling plots surrounding the map is impossible to read and should be larger. To save space you may consider to axis labels only on one plot (assuming it is equal for all), and then minimise the white space.

- Figure 10 caption: replace "all SREX regions" with "each SREX region" – I assume this is what you actually wanted to say (having an average for each region rather than one average over all)?

---

## Author Comment (AC1) · 12 Sep 2017

Dear reviewer,

thank you very much for the helpful comments which will substantially improve the manuscript. We will address all your comments in detail in our final response and focus on the major issues in this response.

An initial response regarding the impact of biases in the atmospheric moisture budget: Based on the work of Liepert and Lo (ERL, 2013) in which they update their previous work (Liepert and Lo, ERL, 2012) for all CMIP5 models, we identified only MIROC5

among our subset of models that potentially exhibits a large drift. This probably further applies to FGOALS-g2, even though Liepert and Lo (2013) used FGOALS-s2. From our own experience we are further aware that the IPSL model is associated with a drying bias over land areas. In order to assess the impact of these models on our results we will perform an additional sensitivity analyses by excluding these models. We totally agree that the identified drift is of great importance and potentially induces spurious changes in hydroclimatological storage components over long time scales. However, the global mean changes identified in Liepert and Lo (ERL, 2013) are equivalent to a maximum of only ca. 0.02mm/day. We further assess the multimodel ensemble in a probabilistic approach, providing median estimates and quantiles, thereby following the recommendation provided by Liepert and Lo (avoiding the ensemble mean). Hence, potentially biased models within the ensemble will not affect the median response provided here.

Regarding the different numbers of available models between scenarios: We will add supplementary material analysing the RCP2.6, RCP4.5 and RCP8.5 scenarios using only those models available in RCP6.0.

As requested in several of the minor comments, we will extend the discussion of several issues throughout the manuscript and all minor corrections and typos will be addressed in the final response. Thank you!

---

## Author Comment (AC2) · 12 Sep 2017

Dear reviewer,

thank you very much for the positive evaluation of the manuscript and your comments, which will help to improve the article. We will address all your comments in detail in our final response and focus on the major issues in this comment.

Regarding the question why we omit regions with P-E<0: Since we focus on global land, we omit locations where P-E<0, since such conditions are generally not present over land at yearly or longer time scales. However, in order to also represent the

scaling over oceans, we will reproduce Figure 2 as a supplementary figure for oceans only. We will now further mention throughout the manuscript that our main focus is on global land areas.

The reviewer asks for a more appropriate way to visualize the regression slopes and their uncertainty: In all dP versus T plots the main assumption is that P is known in case global mean temperature dT=0. We understand that this might be unrealistic. However, we focus on the relative changes in P with changes in T and our approach provides an option to illustrate the uncertainty distribution as a function of temperature change. The violin plot nicely illustrates the uncertainty distribution basically for dT=1K, whereas the dP vs. dT plots illustrate the uncertainty distribution for every dT between 0K and 6K, which, in our assessment, makes it easier to assess probalities/risks as a function of dT.

Regarding potential changes in the variance of P: If the variance increases over time, the uncertainty of the sensitivity coefficient (estimated through resampling residuals) consequently also increases. However, this will not necessarily influence the decomposition of the uncertainties unless changes in precipitation variability are different between scenarios or models.

We will further adress all minor corrections and typos in the final response. Thank you!

---

## Author Comment (AC3) · 12 Sep 2017

Dear reviewer,

thank you very much for the helpful comments which will substantially improve the manuscript. We will address all your comments in detail in our final response and focus on the major issues in this response.

The reviewer asks if internal variability estimated from one run (through resampling) is similar to internal variability estimated from different realisations of one model: The only model from the chosen subset available to us providing a sufficient number of

realisations (10 different realisations) was CSIRO-Mk3-6-0. We already produced violin plots for each SREX region comparing the internal variability distributions estimated from (I) the different realisations of CSIRO against (ii) those estimated from the multi-model ensemble (this was not shown). The differences are marginal and we will include these results as a supplementary figure in the final response.

Regarding the potential influence of aerosol concentrations on our results: We are further aware of the potential influence of different aerosol concentrations on mean precipitation. We referenced in particular the work of Pendergrass and Hartmann (GRL, 2012) and Pendergrass et al. (GRL, 2015) to clarify that mean precipitation scaling depends on the emission scenario (whereas the scaling of extreme precipitation is independent of the scenario). We did, however, not explicitly mention that the differences in mean precipitation scaling can be attributed to differences in the prevailing aerosol concentration, but will do so in the revised version of the manuscript. We will further discuss the scenario uncertainty also in the context of these studies. Nonetheless, quantitatively assessing the extent to which the scenario-specific differences in aerosols relate to the scenario differences in P and P-E requires additional work that goes beyond the rather simplistic approach to attribute relative uncertainty contributions that is used here.

Regarding the linearity assumption: In the final response we will provide supplementary information testing characteristics of the residuals as e.g. autocorrelation and we will provide residual plots for all SREX region allowing for a visual inspection of the linearity assumption. However, if the linearity assumption is not robust due to e.g. heteroscadisity, this results in a higher uncertainty of the linear scaling approach and is therefore accounted for through uncertainty estimation.

Regarding the work of Sippel et al. it is important to mention that our reference period (1980-1999) lies outside the study period (2000-2099) and values from the reference period are hence not used to estimate the scaling factors.

We will further explicitly mention in the revised version of the manuscript that we will focus on global land areas. However, for completeness we will reproduce Figure 2 as a supplementary figure for oceans only.

We will address all other minor corrections and typos in the final response. Thank you!

---

## Author Response (AR1)

This paper applies the regional scaling concept (e.g. Tebaldi and Arblaster, 2014) to the distributions of annual mean precipitation (P ) and precipitation minus evaporation (P-E) anomalies. The rationale behind this methodology is based on the work by Seneviratne et al., 2016, in which it is shown that local temperature and precipitation extremes scale linearly with global mean temperatures in CMIP5 $CO_2$-increasing scenarios projections. The mean and uncertainty ranges of P and P-E local responses scale with the global mean temperature in the same way, independently of the emission scenario. This paper extends Seneviratne et al., 2016, considering annual mean local precipitation response scaling with global mean temperatures over all the SREX regions (Seneviratne et al., 2012). The impacts of uncertainty due to models internal variability, to the inter-model spread and to the scenario spread are also separately accounted for.

**1 GENERAL COMMENTS**

The aim of the paper is focused and clear, and it well suits in the current debate about the regional impact of global temperature change. Exploring the ranges of applicability of the pattern scaling approach allows improving the capability to communicate the impact of climate change to the stakeholders and public opinion. In this sense the assessment of regional changes in P and P-E is of outmost importance for the adaptation to future changes in local water resources.

The regional pattern scaling is here assessed in terms of a basic least squares fit, regressing annual mean P and P-E anomalies over annual mean global temperature anomalies at every grid-point. The uncertainties related to internal variability, model and scenario-related uncertainties for each model are obtained by resampling the residuals 1000-times over each gridpoint. The empirical probability distribution provides thus a way to characterize the range encompassing the median values of the regression slopes, allowing the distinction between very likely (90-100%), likely (66-100%) increase/decrease in P or P-E. The use of a basic linear scaling is justified by the lack of a-priori information about the shape of the annual mean P and P-E distributions over the various regions is not available. This shifts the focus from the choice of suitable downscaling techniques to the evaluation of uncertainty ranges attributable to the regression coefficients. In this respect, I think that the manuscript partially fails in discussing the impact of models' choice. Seneviratne et al., 2016 outlined limitations to the regional scaling pattern approach in this context. Particularly, point 4) of their discussion emphasized the risk of common biases through models for some regional phenomena. They point out that a careful model evaluation against appropriate observations would be necessary to deal with this problem. The internal variability of considered model and the multi-model ensemble uncertainty is addressed in the manuscript, but some more effort should be devoted to the evaluation of each model. Particularly, the biases induced by the imbalance in the water mass budget and the impact of different choices of the model ensembles should be carefully addressed, in order to assess the applicability of the method and the robustness of the findings.

We thank the reviewer for his/her thorough comments, which will help to improve the manuscript substantially. Based on his/her comments, we will provide additional material/figures to support our model choice. We will address all comments in a detailed point-by-point response below.

**2 SPECIFIC COMMENTS**

l. 22-23 p. 3: if not argued the choice of the models may look a bit arbitrary. On one hand the authors rely on Fischer et al., 2014 to choose only one model for each modelling centre. On the Env. Res. Lett.) and this may in principle prevent from consistent estimates of regional changes in P and P-E. Evaluating the long-term mean atmospheric moisture budget in control runs, identifying the regions where climate models diverge from available observations is thus a pre-requisite to this analysis. An inconsistent global mean moisture budget is a potential source of biases and the

impact of adding/removing individual models should be carefully evaluated. In the framework of the regional pattern scaling, it would also be relevant to compare the atmospheric moisture budget separately over continents and oceans with the total runoff from the continents (which is a standard output in climate models, if I am not wrong is named as "mrro"), in order to provide a complete description of the hydrological cycle consistency in the model; other hand they do not consider the impact of biases in the atmospheric moisture budget. Models are known to show diverse estimates of the global mean water budget (cfr. Liepert and Lo, 2012,

As already stated in our initial response, based on the work of Liepert and Lo (ERL, 2013) in which they update their previous work (Liepert and Lo, ERL, 2012) for all CMIP5 models, we identified only MIROC5 among our subset of models that potentially exhibits a large drift. This probably further applies to FGOALS-g2, even though Liepert and Lo (2013) used FGOALS-s2. From our own experience, we are further aware that the IPSL model is associated with a drying bias over land areas. In order to assess the impact of these models on our results, we have added additional figures to the Supplementary showing the results of Figs. 5 and 6 after excluding these models. It is our assessment, that there are qualitatively no substantial differences between the different ensembles that would alter our overall conclusions.
We further agree that the identified drift is of great importance and potentially induces spurious changes in hydroclimatological storage components over long time scales. However, the global mean changes identified in Liepert and Lo (ERL, 2013) are equivalent to a maximum of only ca. 0.02mm/day. We further assess the multimodel ensemble in a probabilistic approach, providing median estimates and quantiles, thereby following the recommendation provided by Liepert and Lo (i.e.~avoiding the ensemble mean). Hence, potentially biased models within the ensemble will not substantially affect the median response provided here.

l. 23-24 p. 3: as also mentioned in l. 23-24 p. 6 the choice of ensembles with different numerosity is inherently a considerable source of uncertainty, unless one considers the 14-member and 7 (in the case of RCP6.0) and 11 (in the case of RCP2.6) members ensembles having the same statistical properties. For the same reasons motivating the previous comments, the impact of adding/removing a model from the ensembles should be carefully evaluated. To be on the safe side, I would suggest to reconsider the RCP2.6, RCP4.5 and RCP8.5 scenarios only using those models that are available in the RCP6.0 and discuss about the presence/absence of significance differences in the results. In the occurrence of significant differences I would try to identify and describe those models significantly reshaping the ensemble distribution;

We added supplementary figures showing the results of Figs. 5 and 6 after excluding all models that are not available in RCP6.0. It is here also our assessment, that there are qualitatively no substantial differences between the different ensembles that would alter our overall conclusions. We, therefore, assume that all ensembles (14-member – RCP4.5, RCP8.5; 11-member – RCP2.6; 7-member – RCP6) have similar statistical characteristics. However, please note that all results in the main text are already provided separately for the individual emission scenarios (Figs. 5,6,7,8).

l. 12-18 p. 4: the definition of variances might be clarified by labelling each sigma with a different subscript, either referring to internal variability, model uncertainty, scenario uncertainty;

Changed.

l. 15 p. 4: following above comment, it should be specified how to deal with the model uncertainty when the ensemble numerosity is lower than 14, e.g. in the RCP6.0 n=7?

See above.

l. 30-31 p. 4: to me it is not clear how the authors deal with uncertainty ranges including the zero value for the slope. Could you please expand this statement?

We rephrased this part and hope it is more understandable now.

l. 5-6 p. 5: the authors might want to comment on the fact that spatial averaging over northern high latitudes is not the same as spatial averaging at lower latitudes, and this shall be considered when discussing the significance of results at different latitudes. I wonder if one could compare circles of latitude somewhat weighting the likelihood of the changes with the cosine of latitude or the surface area covered by each circle.

It is not completely clear to us, what the reviewer requests us to do here. However, we can assure the reviewer that the spatial averages presented in this work are weighted by latitudes. Regression slopes are, however, computed at grid point scale, i.e. that the scaling coefficient at low latitudes represents a larger area than those at high latitudes. We added this information to Sec. 2. Regression slopes for the SREX regions are computed by using weighted averages of P and P-E from the particular regions.

l. 19-22 p. 6: the authors mention the different shapes of the uncertainty distributions for different SREX regions in P and P-E regression slopes. Could you please specify whether you refer to the P, P-E or both variables. Otherwise these statements appear a bit arbitrary and one might want to consider removing them;

We changed this and now clearly state, separately for P and P-E, which regions are affected.

l.1 p. 7 (and l. 25 p. 8): please specify the meaning of "significantly";

We rephrased this part and now specifically explain what we mean here.

l. 4-6 p. 7: the authors list here a number of SREX regions characterized by larger/smaller internal variability, model uncertainty, scenario uncertainty compared to other regions. I think some more explanation might be welcomed here, rather than just listing the findings over the various regions. Why these regions, rather than others? For instance, the large model uncertainty over northern high latitudes might be related to the more relevant signal ("very likely increase" in precipitation), whereas the large internal variability over the two sides of the Tropical-Northern Pacific might reflect some relatively well understood mechanisms of inter-annual variabilities, such as the QBO (cfr. Labat et al., 2004, Geophys. Res.Lett.);

Thanks. We expanded this part and added the reference.

l. 14-16 p. 7: repetition of l. 2-4 p. 3, consider removing;

We rephrased this part to avoid any repetition.

3 TECHNICAL COMMENTS

l. 25 p. 3: remove one "in";
l. 17-18 p. 4: replace "coefficient" with "coefficients";
l.23 p. 6: replace "causes" with "cause";

l. 14 p. 8: replace "extent" with "extend;
Table 2: the acronym for Northern Australia should be NAU (instead of NAS);

Thanks! Changed.

In this manuscript, Greve, Gudmundsson, and Seneviratne examine the scaling of local and regional precipitation and P-E with global mean surface temperature in climate change projections. They diagnose the likelihood of increases or decreases with warming in both quantities, and characterize and identify uncertainty due to internal variability, structural model differences, and differences in emissions scenario. To address the impacts of P and P-E on the 1.5 and 2C warming targets, they quantify the P and P-E responses and their uncertainty in each of a variety of land regions in response to the two targets. They find that the mean changes in P and P-E are indistinguishable for 1.5 and 2C, but that the two warming targets do differ in the tail of risk estimates, with a higher risk of the largest changes for 2C warming compared to 1.5C

This work makes a useful contribution to the literature, as regional changes in mean precipitation scaling have not yet been diagnosed. The maps and violin plots for individual regions are particularly useful. There are a few issues I think should be addressed to improve the manuscript.

We sincerely thank the reviewer for the positive evaluation of the manuscript and his/her comments, which substantially helped to improve the manuscript. We provide a detailed point-by-point response below.

Scientific issues

P3 line 26-27: Why omit locations where P-E<0?

We here focus on global land. Locations where P-E<0 are omitted since such conditions are generally not present over land at yearly or longer time scales. We will now further mention throughout the manuscript that our focus is on global land areas. Hence, we also decided against providing maps for ocean regions, since the underlying mechanisms and drivers are potentially very different from those over land areas. It is now our assessment that this should be analyzed individually and properly in future studies.

Figures 1, 3, and 4: In all dP versus T plots with the exception of the top panel of Fig. 1, the regressions cross through the origin. The uncertainty in the regression slope is shown as occurring entirely at the upper end of the temperature change axis. These are in conflict with the top panel of Fig. 1, where the regression slopes do not pass through the origin. Internal variability is always present, so we would expect small changes in dP even when dT=0. Is there a better way to visualize the range of regression slopes and their uncertainty? The violin plots are quite useful and do not contain these distortions.

As already stated in our initial response, in all dP versus T plots the main assumption is that (initial) P is known in case global mean temperature change dT=0. We understand that this might be unrealistic. However, we focus on the relative changes in P (dP) with changes in T, such that dP=0 when dT=0 and hence, the lines cross the origin. This approach provides an option to illustrate the uncertainty distribution as a function of temperature change. The violin plot nicely illustrates the uncertainty distribution basically for dT=1K, whereas the dP vs. dT plots illustrate the uncertainty distribution for every dT between 0K and 6K, which, in our assessment, makes it easier to assess probabilities/risks as a function of dT.

P4 line 28/30: I believe the 10th-90th percentile confidence corresponds to p=0.2, rather than p=0.1. In addition, why do you choose 10th and 90th percentile – since these are wider bounds than is customary? Why not 5 and 95 (p=0.1), or 2.5 and 97.5

(p=0.05)?

Thanks! We rephrased this part to also emphasize why we chose the 10th and 90th percentile. If the zero coefficient is within the range, it means that the probability of experiencing a scaling response of different sign compared to the median response, is, at least, 10% or higher. This corresponds to the "very likely" definition of the IPCC, which is used throughout the manuscript.

P4 line 10, P6 line 28: The methodology of Hawkins and Sutton (2009) assumes that variance is constant over the course of simulations. They only examined temperature, for which this assumption is more or less valid. It seems to me that resampling residuals would rely on the same assumption. For precipitation, it is not the case that precipitation variability is generally constant – instead, it increases in most regions (e.g., Räisänen, 2002). Do you think increasing precipitation variability would affect your uncertainty decomposition, and if so, how?

As already stated in the initial response, if the variance increases over time, the uncertainty of the sensitivity coefficient (estimated through resampling residuals) consequently also increases. However, this will not necessarily influence the decomposition of the uncertainties unless changes in precipitation variability are different between scenarios or models.

Typos and grammatical comments

P2 line 31: "comprehensive subset": This is contradictory, since a subset is by definition not comprehensive.

We now use "representative".

P5 line 9: "A very likely decrease is rarely found only in South Africa." I think what you mean is that a decrease with very likely confidence is found only in South Africa, and therefore it is rare; your wording means something else: that very likely decrease is often found in many places, rarely only in South Africa.

Rephrased.

P6 line 12-14: "the higher emission scenarios are usually enclosed by the low emission scenarios and the uncertainty is narrowing down"; "partly huge differences": These phrases are not quite grammatically correct.

We rephrased the sentence.

Fig. 1: "Global mean Temperature" should probably be "Global mean Temperature Change"

Changed.

The authors investigate regional changes in precipitation (P) and water availability (expressed in terms of precipitation minus evaporation, P-E) as a function of global temperature changes in a sub-set of the CMIP5 simulations. They further decompose the uncertainties by sources related to climate variability, scenario, and model choice. They find robust changes towards wetting in northern high-latitude regions, and tendencies towards drying in subtropical regions, however associated with larger uncertainties. In particular, they also discuss changes related to political global warming limits of 1.5K and 2K.

This study is a worthwhile contribution to the literature, addressing the relevant topic of regional impact-relevant responses related to different amounts of global warming. The manuscript is mostly well written, but some clarification is needed at a few places. I also have a few more major questions related to the methodology, but think that it should be possible to clarify these with some revisions.

We sincerely thank the reviewer for the thorough and overall positive evaluation of the manuscript. His/her comments will help to improve the manuscript and we provide a detailed point-by-point response in the following.

Major comments:

(1) The authors use resampling to estimate the effects of internal climate variability. They mention that this leads to similar results as using different realisations of one model but do not show results. As estimation of different uncertainty sources, including variability, is one of the main goals of this paper, I think the authors should provide evidence that their approach by just resampling results from one run does actually lead to comparable results to analysing different runs. This seems important as usually effects of variability are estimated from a number of runs started from different climate states with respect to internal variability.

The only model from the chosen subset that was available to us providing a sufficient number of realisations (10 different realisations) was CSIRO-Mk3-6-0. We have now added plots for P similar to those provided in Figs. 3 for each SREX region comparing the slope estimates from (i) the 10 different realisations of CSIRO-Mk3-6-0 against (ii) those estimated from the resampling approach. These results are now provided as a supplementary figure in the final response. We further performed a Kolmogorov-Smirnov test to determine if the two samples, (i) 10 slope estimates from CSIRO-Mk3-6-0 and (ii) 1000 slope estimates from the resampling, are from the same parent distribution. For each SREX region, the null hypothesis, that the two samples are from the same parent distribution, can not be rejected. Respective p-Values are also provided in the new supplementary figure.

(2) The authors document some larger differences in the response between different scenarios, and seem to discuss these differences in the context of different strength of the GHG forcing. However, also the aerosol concentrations differ between the different RCP scenarios, and I wonder to which extent these scenario differences of P and P-E changes could be attributed to differences in aerosols?

We are aware of the potential influence of different aerosol concentrations on mean precipitation. We referenced in particular the work of Pendergrass and Hartmann (GRL, 2012) and Pendergrass et al. (GRL, 2015) to clarify that mean precipitation scaling depends on the emission scenario (whereas the scaling of extreme precipitation is independent of the scenario). We did, however, not

explicitly mention that the differences in mean precipitation scaling can be attributed to differences in the prevailing aerosol concentration, which is now briefly discussed. Nonetheless, quantitatively assessing the extent to which the scenario-specific differences in aerosols relate to the scenario differences in P and P-E requires additional work that goes beyond the rather simplistic approach that is used here, which aims to attribute relative uncertainty contributions from different sources.

Specific and technical comments:

- Abstract, line 3: I'd remove "large" as I don't think 14 model simulations is a "large" sub set of the total number of runs available in CMIP5

We romved "large" here.

- Abstract, line 6: (Please also check throughout the text!) I suggest avoiding "dependency" when discussing the relationship of regional climate with global mean temperature. Better just say "linear relationship" here.

We now avoid the word "dependency" throughout the manuscript.

- page 1, line 21: suggest adding "public and political debate"

Changed.

- page 2, line 14: I wonder if the assumption of a linear relationship is justified when investigating changes at individual grid cells from individual ensemble members. Especially P and P-E can be rather noisy variables, strongly affected by low-frequency variability, so it might help to justify the robustness of the approach if the authors provided some tests whether linearity is a reasonable assumption in this context.

The linearity assumption is potentially valid if e.g. the residuals are (i) normally distributed and (ii) not autocorrelated. We now provide supplementary information on these statistical properties. We show for each gridpoint the number of models for which (i) the Kolmogorov-Smirnov test does not reject the null hypothesis (i.e. residuals are normally distributed) and (ii) there is no significant lag1 autocorrelation of the residuals. However, it is important to note that a further and more thorough assessment of the linearity assumption is aggravated through the large amount of data (which does e.g. not allow for any visual inspection at grid point scale, however, we added this for the SREX regions).

Nonetheless, in this study we aim to assess the scaling relationship between dT and dP or d(P-E) in terms of a single number (the scaling coefficient), which might be expressed through the slope of the regression line between the two variables. This is a simple approach, not accounting for any nonlinearities, which would ultimately lead to a more complex scaling relationship (not a single number, but potentially a more complex function). In this sense, if the linear assumption is valid, the slope estimate itself is a good representation (or a good model) of the relationship, if the linear assumption does not apply, the slope provides a bad representation (or a bad model) of the relationship. However, we do not provide deterministic estimates of the scaling coefficient alone, but we also thoroughly assess the uncertainty of the slope through resampling the residuals and repeating the regression analysis (1000 times at each grid point for each model and for each scenario). This provides us with an uncertainty estimate (even an uncertainty distribution) of the slope estimate that corresponds to the validity of linearity. In regions with high uncertainties related

to the slope estimates (evidently in many arid and hyper-arid regions), both the tests for normality and autocorrelation fail for most models. In most world regions, however, residuals are normally distributed and not autocorrelated for the majority of models. This is now shown in a supplementary figure.

- page 2, line 20: This sentence is ambiguous, it seems like "this work" and "this analysis" refer to different studies, but it is not really clear what refers to what.

We rephrased the sentence.

- page 3, line 1: insert "global" warming-degree targets

Changed.

- page 3, line 23: many readers may not be aware what exactly the "prerequisites provided in Fischer et al." are – for better readability please briefly summarise

We added this information.

- page 3, line 25: the historical runs include the year 2005, therefore if starting in 1980 this should be "26 years" and "from 2006 onwards".
- page 3, line 26: remove duplicate word "in"

Changed.

- page 3, line 26/27:  Sippel et al (https://doi.org/10.5194/hess-21-441-2017) discuss that assessing changes relative to a short reference period may lead to bias in the out-of-reference period. As the authors chose here to use only 20 years as baseline, I am wondering whether their quantifications of changes would be affected by such biases?

Regarding the work of Sippel et al. it is important to mention that our reference period (1980-1999) lies outside the study period (2000-2099) and values from the reference period are hence not used to estimate the scaling factors.

- page 3, line 27:  Sentence not clear, does "majority of models" suggest that some models are treated different than others?

We rephrased this sentence.

- page 4, line 1: What kind of least squares fit did you use, e.g. ordinary or orthogonal (i.e. minimising squared differences only in y-direction or in both x and y-directions)? I think there may also be some error in the T values, so orthogonal least squares might be most appropriate?

We use here ordinary least squares and mention this in the text now. Since we want to focus on uncertainties in P and P-E, we decided against using orthogonal squares. Nonetheless, you are right and there might be also some errors in dT, but it is our assessment that these errors are not necessarily well captured through using orthogonal least squares.

- page 4, line 24: Please check, is it 10th and 90th quantile, or 25th and 75th quantile as written on Figure 2?

We now provide the correct figure showing the 10th and 90th quantile.

- page 4, line 25: Very confusing use of parentheses. If I follow your logic that the text in parentheses indicates some opposite results/statements then this sentence seems to suggest negative values always relate to (P-E) – which of cause is nonsense. Please also see this text by A. Robock (https://eos.org/opinions/parentheses-are-are-not-for-references-and-clarification-saving-space), and consider rewriting this sentence in a more readable (and clearer!) way.

You are right.We removed this sentence and rephrased this part. Thanks!

- page 5, line 5: based on only 14 models, the 90-100

We are not sure what the reviewer intends us to do here? The "very likely" decrease presented here is not just based on 14 models, but on the uncertainty distribution consisting of several thousand slope estimates that were computed through resampling the residuals.

- Page 5, line 21: Figure 4 shows the scaling relationships as regression lines, it does not explicitly show the "coefficients" as claimed in this sentence.

We changed this. However,  please note that in all dP versus T plots the main assumption is that (initial) P is known in case global mean temperature change dT=0, which might be unrealistic. However, we focus on the relative changes in P (dP) with dT, such that dP=0 when dT=0 and hence, the lines cross the orign. This approach provides an option to illustrate the uncertainty distribution as a function of temperature change. The dP vs. dT plots provided here illustrate the uncertainty distribution for every dT between 0K and 6K, which, in our assessment, makes it easier to assess probalities/risks as a function of dT.

- page 5, line 26: replace "/" by "or"
- page 6, line 1: remove "much"
- page 6, line 3: replace "within" with "in", and "parts" with "individual grid cells"
- page 6, line 4: replace "within" with "in"
- page 6,  line 7:  consider adding the clarification "(very) likely decrease [across all scenarios]".
- page 6, line 21: replace "many" with "some"

Changed. Thanks!

- page 6, line 25:  The structure of the Results section seems somewhat confusing. Section 3.2 is named "sources of uncertainty" – but didn't already section 3.1 discuss one specific source of uncertainty? Please consider restructuring Section 3 more logically, or at least use more suitable sub-section names, e.g.  "3.2 Comparing different sources of uncertainty".

We now introduced different sub-section titles.

- page 7,  line 26:  add "with stronger global warming" (or similar) at the end of the sentence after "P-E".

Added.

- page 9,  line 1:  This sentence is a literal repetition of page 7,  line 30-32.   Please

consider rephrasing one of these instances. Otherwise this is a very nice conclusion.

Thanks! We rephrased the part on page 7.

- Figure 2: It looks like ocean areas and Antarctica were removed. This is not explicitly mentioned in the text – are these regions removed due to P-E<0 here (page 3, line 27) ?  As mentioned above, please also check for consistency whether it is 10th/90th percentile (or if you wish to express in quantiles: 0.1 and 0.9), or 25th/75th

We now provide the correct figure showing the 10th and 90th quantile. We will now further mention throughout the manuscript that our focus is on global land areas. Hence, we also decided against providing maps for ocean regions, since the underlying mechanisms and drivers are potentially very different from those over land areas. It is our assessment that this should be analyzed individually in future studies.

- Figure 3/4:  The text and labelling of the T-P scaling plots surrounding the map is impossible to read and should be larger. To save space you may consider to axis labels only on one plot (assuming it is equal for all), and then minimise the white space.

Changed.

- Figure 10 caption:  replace "all SREX regions" with "each SREX region" – I assume this is what you actually wanted to say (having an average for each region rather than one average over all)?

Changed. Also in the other figure captions.

[revised manuscript text omitted]

---

## Referee Report (RR1)

The current version of the manuscript thoroughly addresses the concerns raised by me and the other reviewers. In particular, the authors have discussed the choice of the models from the point of view of the moisture budget and the relevance of having ensembles with different number of members. The authors also test the validity of the Gaussian distribution assumption, thus significantly improving the statistical robustness of the methodology. For these reasons, I would recommend the acceptance as is.

---

## Author Response (AR2)

The authors have satisfactorily revised the manuscript in response to the reviewers' comments. Below are a few additional suggestions to ensure clarity of the manuscript.

We sincerely thank the reviewer for his/her positive evaluation of the manuscript. The final comments provided below have been very useful in order to increase the readibility of the manuscript.

Throughout the manuscript, the authors refer to supplementary information for additional material. However, I find the general reference to a document with eight figures, leaving it to the reader to find the relevant information themselves, is not particularly helpful. Readability could be much improved by pointing the reader to the specific supplementary figure that is relevant in the context of the respective claims in the main text (e.g. page 4 line 11, page 5 line 34, page 6 line 19, and check throughout the manuscript so that all supplementary figures are referenced in the text).

We now refer specifically to the individual figures in the supplementary information.

Page 4 line 12: change "regions" to "grid cells"

Changed.

Page 4 line 17: Please be clear about what this failing of the tests in hyper-arid regions means for your study, and how do you deal with this in your analysis

We now clearly state that linear scaling might not be the most appropriate approach to assess P and P-E sensitivities to climate change in arid to hyper arid regions and that this potentially causes spurious results in these regions

Page 4 line 30: change "P (P-E)" to "P and P-E"

Changed.

Page 5 line 7: delete "at least" (you say "or higher" which means the same)

Changed.

Page 5 line 18: Please be exact in your language. I do not think that the "scaling coefficient" is a function of T because the coefficient is derived from regression of P values across the entire T range, so should be constant?
Page 5 line 29: same for scaling coefficient of P-E

Changed.

page 6 line 20: change "P (P-E)" to "P and P-E"

Changed.

page 6 line 21: change "(Fig. 8)" to "and Figure 8, respectively" or similar

Changed.

page 6 line 24: please specify "more certain signal for higher emissions"; change "huge" to "large"

Changed.

page 6 line 29: change "P (P-E)" to "P and P-E"

Changed.

page 7 line 1: please specify differences IN THE DISTRIBUTION SHAPES, or similar

Changed.

page 7 line 3: I still don't think Sources of uncertainty is a particular useful heading for section 3.2 given also 3.1 discusses a source of uncertainty. Please consider a more useful structuring or heading, e.g. an alternative heading for 3.2 could be "Comparing different sources of uncertainty".

Changed.

Page 7 line 16: I do not follow the discussion that the large uncertainty from model choice should be related to certainty in the scaling response, in fact this seems illogical. Can you please better explain this relation?

We removed this confusing sentence. Thanks.

Page 7 line 19: Also I do not see how QBO (acting on inter-annual time scales) would affect long-term climate change scaling rates – please explain.

We actually also removed this part. This was suggested by another reviewer, but we also do not strongly agree with this statement nor think it is of great relevance here.

Page 8 line 8: this sentence seems to mix up singular and plural: "difference...is" or "differences...are"?

Changed.

Page 8 line 11: change "of more" to "by more"

Changed.

page 8 lines 20 and 21: change "P (P-E)" to "P and P-E"

Changed.

page 9 line 14: why "naturally" ?

Removed.

Figure 9: The authors should specify here for which temperature change or for which time scale these contributions of uncertainty sources are valid. For example the role of internal variability would depend on time scale (Rowan and Sutton 2009).

This relates to the scaling coefficient itself (hence for a change in T of 1K).

[revised manuscript text omitted]